# Investigating the Multi-Recyclability of Recycled Plastic-Modified Asphalt Mixtures

**Gaetano Di Mino** [1] , **Vineesh Vijayan** [1,*] , **Shahin Eskandarsefat** [2] , **Loretta Venturini** [2] and **Konstantinos Mantalovas** [1]

1   Department of Engineering, University of Palermo, Viale delle Scienze, Edificio 8, 90128 Palermo, Italy; gaetano.dimino@unipa.it (G.D.M.); konstantinos.mantalovas@unipa.it (K.M.)
2   Iterchimica S.p.A, Via Guglielmo Marconi 21, 24040 Suisio, Italy; shahin.eskandarsefat@iterchimica.it (S.E.); loretta.venturini@iterchimica.it (L.V.)
*   Correspondence: vineesh.vijayan@unipa.it

**Abstract:** Although the benefits of asphalt recycling have been scientifically proven and several best practices are being implemented, further research is required in specific and specialized areas. One of these circumstances is the recycling of Reclaimed Asphalt Pavements (RAPs) that contain asphalt modifiers such as elastomers and/or plastomers. Following the principles of the circular economy and considering the sustainability implications of asphalt mixtures, this paper deals with the multi-recyclability of asphalt mixtures containing 50% RAP with and without a recycled plastic asphalt modifier and rejuvenating agent. The recycled plastic asphalt modifier was made of hard recycled plastics and was introduced to the mixture via a dry method. The research focuses on the characterization of binders via conventional, rheological, and chemical analysis. To control the consistency and variables of the mixtures, the RAP was produced artificially in the laboratory following an ageing protocol for loose asphalt mixtures. According to the obtained results, at all three cycles of binder recycling, comparable properties for (i) the extracted binders from the recycled plastic-modified asphalt mixture, (ii) the extracted binders from the control un-modified mixture, and (iii) the reference bitumen 50/70 were obtained. This was even noticed when a nearly similar quantity of the rejuvenator was needed during the rejuvenator optimization process. Overall, it can be deduced that from the binder-scale point of view, the mixture containing the introduced recycled plastic additive could be recycled for multiple life cycles without any degradation of its mechanical and physical properties.

**Keywords:** asphalt recycling; multi-recycling; recycled plastic asphalt modifier; dry method; loose asphalt mixture ageing; circular economy; sustainability

## 1. Introduction

### 1.1. State-of-the-Knowledge

Asphalt mixtures are considered the most widely used paving components in the road construction industry considering their benefits of cost efficiency, noise reduction, durability, and ride comfort [1]. However, as a key player in the road engineering sector, the production of asphalt mixtures significantly impacts the environment by exploiting natural resources, emitting greenhouse gases, and generating massive waste. In addition to these environmental challenges, the increasing resource scarcity and the rising cost of crude oil have forced the road industry to focus on the implementation of sustainability and circular economy principles [2–5]. Asphalt recycling is considered an effective solution for embedding sustainability in the asphalt industry, which reuses/recycles valuable materials to cope with both environmental and economic challenges. The use of reclaimed asphalt pavement (RAP) has been proven a promising solution in asphalt production that allows for minimizing the consumption of non-renewable resources, waste generation, environmental

pollution, and the cost of production [6–8]. Nevertheless, the amount of RAP incorporation in new asphalt mixture is still limited for several reasons such as technical issues related to production, performance properties, and durability. The issues are mainly related to the presence of aged binder in the RAP that increases the stiffness of the resulting asphalt mixture and increases the cracking susceptibility of the pavement. Many studies have reported that the use of rejuvenators can significantly increase the reusing/recycling of RAP with performance properties similar to or even superior to that of reference asphalt mixture without RAP [9–11].

Anyway, the mentioned environmental and economic challenges, the unpredictable climatic condition, and the rapid growth in the traffic volume have impacted the performance and service life of asphalt pavements. These extreme conditions have steered to premature pavement distresses such as rutting, fatigue cracking, and thermal cracking [12]. Hence, to tackle such technical issues related to performance properties, as one solution, polymer modification has been practiced since the 1980s. The polymer incorporated in the asphalt modification can be categorized mainly into plastomers and elastomers. The plastomers such as polyethylene (PE), polypropylene (PP), and ethylene-vinyl acetate (EVA) and the elastomers such as styrene-butadiene-styrene (SBS) and styrene-butadiene-rubber (SBR) are the most commonly used polymers. Depending on the type and mode of use, the use of polymers could enhance either the rheological properties of the asphalt binder or the mechanical and performance properties of the asphalt mixtures [13,14]. There are two methods to incorporate polymers in asphalt mixtures: the wet method, and the dry method. In the wet method, polymers are introduced into the bitumen and blended using a mechanical mixer producing polymer-modified bitumen (PMB). On the other hand, in the dry method, the polymer is directly added to the hot aggregates before mixing it with the bitumen during the mixture production. Depending on the type of the polymer, in this case, the polymer could be considered as an alternative to aggregate, filler, or an asphalt binder modifier improving its properties [15].

Within the context of sustainability and asphalt modifiers, the growing concern over plastic waste around the world has made researchers focus on investigating the viability of utilizing recycled waste plastics as an asphalt modifier. This could also help to manage the accumulation of plastic waste in a sustainable way. The literature shows that the incorporation of some recycled plastics could enable promising results as modifiers for asphalt binders or mixtures [16]. It has been reported that the addition of some of the recycled plastic modifiers improves the performance-related properties of asphalt mixtures such as resistance to moisture and rutting [17]. In this respect, a study conducted by Poulikakos et al. has reported that the dry addition of 1.5% polyethylene (PE) waste (on the weight of asphalt mixture) in asphalt mixture showed improved resistance to permanent deformation, higher elastic behaviour, low creep rate, and higher creep modulus. However, the increase in the percentage of PE to 5% exhibited an increase in permanent deformation [18]. The increased stiffness and resistance to permanent deformation have been studied both for the mixtures modified via wet and dry methods. D' Angelo et al., studied the performance of asphalt mixtures modified using the wet method with two types of recycled plastics. The outcome of their research demonstrated that the plastic-modified binder blends performed better than the neat bitumen at high temperatures during the rheological analysis which shows an improved rutting resistance. They also concluded that the modified blends exhibited comparable results in terms of bond strength [14]. Nevertheless, the asphalt binder/mixture modification via recycled plastics has faced some challenges. The growing trend of asphalt modification with recycled plastic modifiers raised concerns related to its recyclability. While the mechanical and performance properties of asphalt mixtures modified with recycled plastics are very clear at medium and high temperatures, the properties at low temperatures and fatigue are less investigated. Few available research on the fatigue and low-temperature properties demonstrate that a similar or slightly superior performance compared to reference asphalt mixtures is obtained when the mixture is modified by recycled plastics [19].

From another point of view, considering the Life Cycle Assessment (LCA), the RAP containing the recycled plastic modifier in the previous cycles was also a matter investigated by some researchers. A study conducted by Lu et al. concluded that the studied recycled plastic-modified asphalt mixtures are recyclable and provide similar properties compared with the reference (non-modified) RAP. According to this study, the incorporation of 30% RAP-containing recycled plastic in new asphalt mixtures did not have any adverse effects on their cracking and rutting performances [20].

### 1.2. Objective

While the technical performance of recycled plastic asphalt binders and mixtures has been researched and shown in several studies, there are still some aspects that have been less focused upon. The properties of RAP containing recycled plastics, and the recyclability of such RAPs are some of the missing information that need addressing. Considering the increasing tendency of high RAP reusing/recycling a clear understanding of RAP-containing recycled plastic modifiers is essential. Currently, there are limited studies on the topic and therefore this research aims at investigating the characterization and viability of multi-recycling of recycled plastic-modified asphalt through a binder-scale study.

## 2. Materials

In this study, limestone aggregates were used to produce a dense graded asphalt mixture for a wearing course. The design gradation used in this study was in accordance with an Italian technical specification for the highways [21]. As for the asphalt binder, a paving-grade bitumen 50/70 was used in the study. The bitumen was selected based on firstly, the local availability and secondly, as the commonly used binder in the production of asphalt mixtures for the highways of the Sicily region in Italy.

Considering the main objective of this paper, a recycled plastic asphalt modifier was used for the modification of the asphalt mixture. The additive shown in Figure 1 was made of recycled hard plastics and graphene nanomaterials so-called Graphene-enhanced Recycled-Plastic additive, hereinafter (GRP). Some of the given technical properties are summarized in Table 1. It is worth mentioning that the waste plastics used to produce this asphalt modifier are hard plastics and not softer ones such as plastic bags and water bottles.

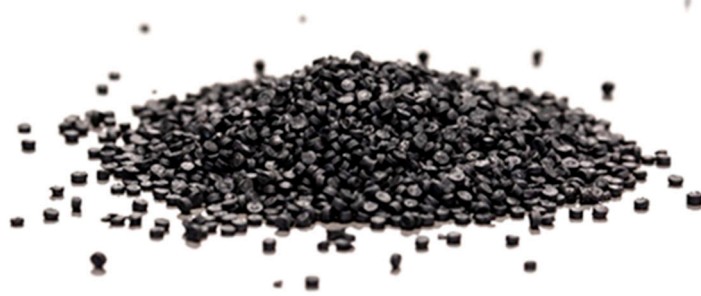

**Figure 1.** Graphene-enhanced Recycled-Plastic (GRP) asphalt modifier.

**Table 1.** Some of the given technical properties of recycled plastic-modifier.

| Properties | Unit | Value/Aspect |
|---|---|---|
| Material State | - | Granules |
| Colour | - | Black |
| Apparent density at 25 °C | g/cm$^3$ | 0.4–0.6 |
| Softening Point | °C | 160–180 |

In addition to the asphalt modifier and considering the multiple recyclability of the mixtures with high content of RAP, a bio-based rejuvenator; hereinafter (R) was also used as the recycling agent in the research. The choice of the rejuvenator was based on material screening according to the proven performance of the commercially available rejuvenators in the market. Among many available rejuvenators, the quality and performance of this rejuvenator have been examined in several broad research works [22,23] that ascertained the choice of it. Some of the technical properties of the rejuvenator are presented in Table 2.

**Table 2.** Some of the given technical properties of the rejuvenator.

| Properties | Unit | Value/Aspect |
|---|---|---|
| Material State | - | Liquid |
| Colour | - | Dark brown-purple |
| Density at 25 °C | g/cm$^3$ | 0.85–0.95 |
| Viscosity at 25 °C | cP | 50–150 |
| Flash point | °C | ≥200 |

## 3. Methods

### 3.1. Research Plan

Considering the objective, in this research two sets of asphalt mixtures were investigated: (i) un-modified asphalt mixtures as the control mixtures without having any asphalt modifier and just a rejuvenating agent used for the two cycles of recycling, and (ii) recycled-plastic modified asphalt mixtures with a rejuvenating agent. It is worth mentioning that in addition to the binders extracted from the mentioned mixtures, reference bitumen 50/70 at unaged and aged states was also considered during the experimental works and the data analysis. The multi-recyclability of the asphalt mixtures was evaluated by applying a defined research plan, shown in Figure 2. The study aimed at investigating the viability of multi-recycling via binder-scale characterization. This would enable knowledge of aged binder properties through the multi-recycling of asphalt mixtures that play a key role in the circular economy. According to the literature, high RAP incorporation is restricted due to the high stiffness of the aged binder as one of the prime reasons. Within this context, the properties of RAP from recycled plastic-modified asphalt mixtures are still unknown or not yet broadly researched. Responding to the missing knowledge in the circular economy of plastic-modified asphalt mixtures, the study on RAP from such modified asphalt mixtures is inevitable.

The choice of 50% RAP incorporation in the study was based on the foreseen level of recycling in the asphalt industry of Italy and the feasibility of the recycling approach in many existing asphalt plants. Most asphalt plants in Italy are batch plants, where the maximum allowable recycling rate in such asphalt plants is limited to 50%. Moreover, the most practiced method to incorporate RAP in the asphalt plant is by superheating the virgin aggregates, so that when RAP comes in contact with the virgin aggregates during the mixing phase would dry and heat the RAP by conduction and it prevents direct exposure of RAP with the flame to minimize the ageing and the release of emissions. In such case, if the recycling rate exceeds over 50%, the superheating temperature of virgin aggregates needs to be increased which creates a higher cost for the energy and less chance of uniformly conducting the heat to the RAP as the rate of virgin aggregates is lower than the rate of RAP in the mix. Furthermore, according to most of the Italian technical specifications, the current maximum allowable rate of RAP inclusion is 15–20% in the wearing course.

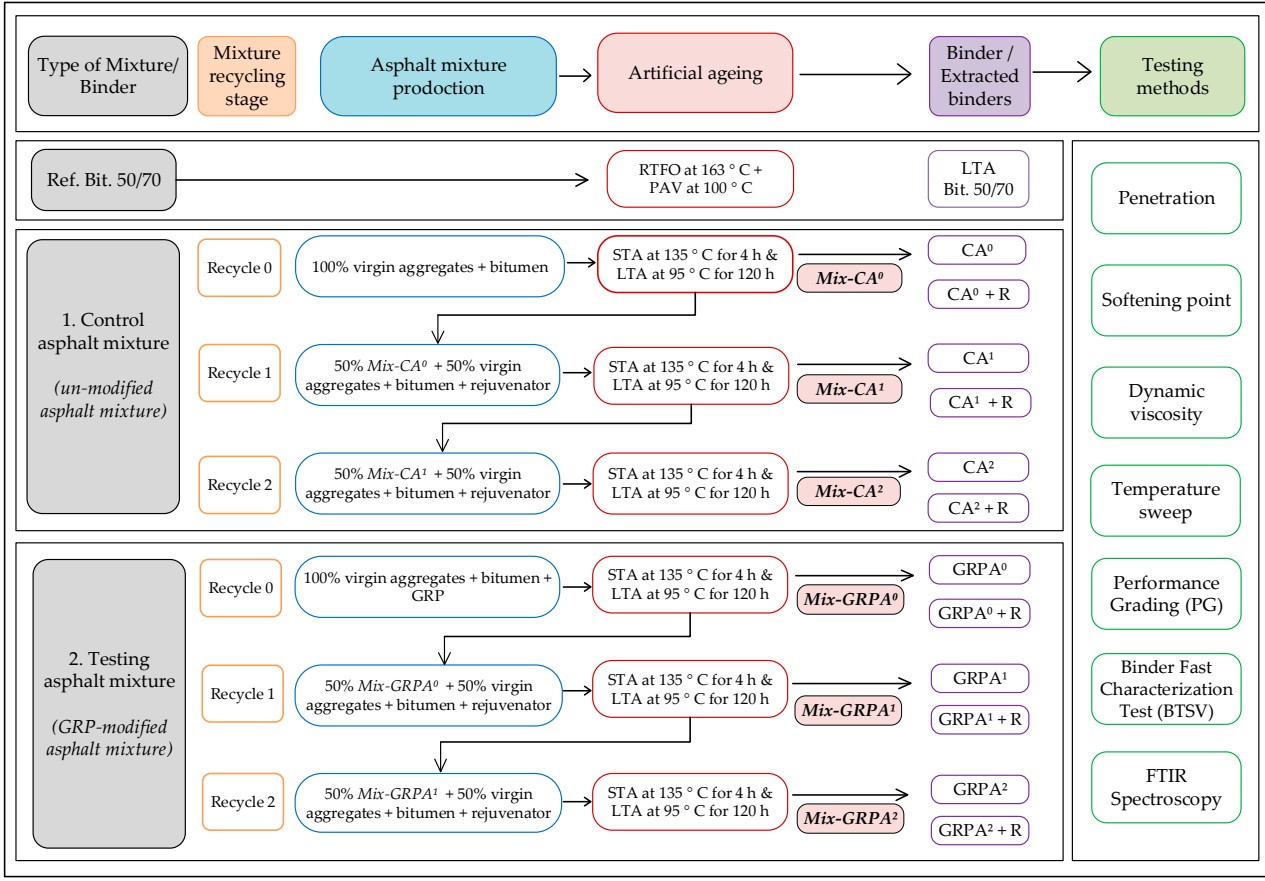

**Legends:**
1. Mix-CA$^i$ : long-term aged Control Asphalt mixture of recycling cycle i; where, i = 0, 1, 2.
2. Mix-GRPA$^i$ : long-term aged GRP-modified Asphalt mixture of recycling cycle i; where, i = 0, 1, 2.
3. CA$^i$ : extracted aged binder from Mix-CA$^i$ where, i = 0, 1, 2.
4. GRPA$^i$ : extracted aged binder from Mix-GRPA$^i$ where, i = 0, 1, 2.
5. RTFO : Rolling Thin Film Oven.
6. PAV : Pressure Ageing Vessel.
7. STA : Short-Term Ageing.
8. LTA : Long-Term Ageing.
9. GRP : Graphene-enhanced Recycled Plastic.
10. R : Rejuvenator.

**Figure 2.** Research plan.

### 3.2. Mix Design and Sample Preparation

The mix design used in the study was according to an Italian technical specification [21] for a wearing course having a maximum aggregate size of 12.5 mm. The grading distribution curve of the designed mix is demonstrated in Figure 3.

The mixture was optimized using volumetric and Marshall methods, where the asphalt binder content of 5% (on the weight of the aggregates) was defined. Having the control asphalt mix optimized, for the testing asphalt mixture containing recycled plastic asphalt modifier, 6% GRP (on the weight of the bitumen) was added to the mixture applying the dry method. It is worth mentioning that the choice of the dosage of the GRP is firstly based on the average value mentioned in recommended Technical Data Sheet (TDS) of the product and recommendation and secondly based on the test results of similar research works where the same additive was used [12,17]. For this purpose, as shown in Figure 4, during the mixing phase, the GRP granules were added to the hot aggregates and maintained in the oven at a temperature of 170 ± 5 °C for approximately 1 h until the granules become

softened, before adding the hot bitumen maintained at 160 °C. In fact, this method was recommended as the mixing procedure in the TDS of the product.

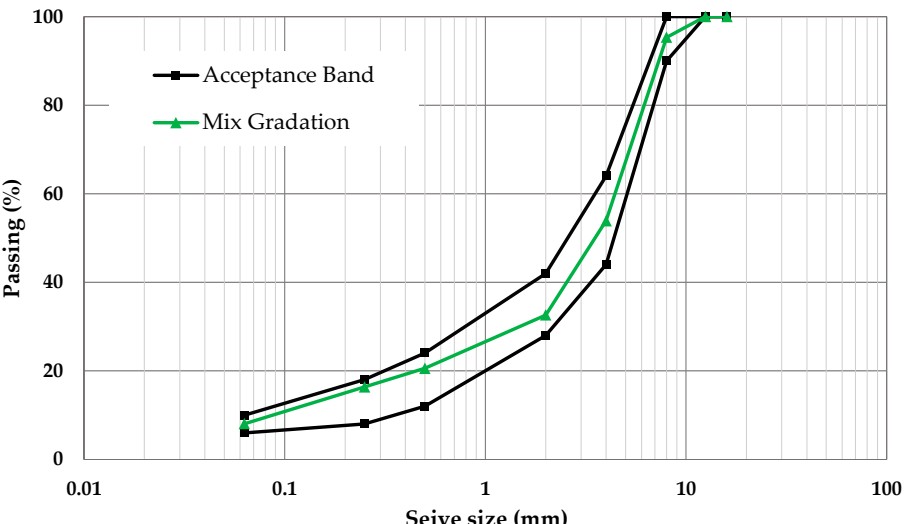

**Figure 3.** Mix design gradation.

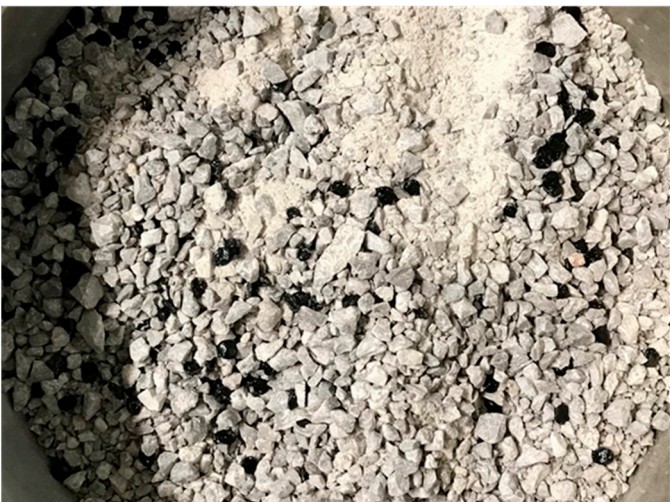

**Figure 4.** The state of the GRP with hot aggregates before the mixing phase.

### 3.3. RAP Manufacturing

Although evaluating the properties of in situ aged RAP is vital in any recycling study, due to the theme of the research, which deals with multiple recycling and controlling the consistency of the RAP, artificial RAP was manufactured. In addition, the introduced GRP modifier is a novel product, and yet the corresponding RAP was not available. Therefore, such modified asphalt mixtures need to be produced in the laboratory to simulate the properties of RAP.

For this purpose, to produce RAP, both sets of asphalt mixtures, for control and testing asphalt mixtures, were artificially aged following a laboratory ageing protocol. For this purpose, a broad investigation was carried out to determine the temperature and duration needed to be applied best simulating the ageing parameters of RAP, considering the RTFO and PAV aged properties of the bitumen used in this study. During this phase, the full rheological properties of the aged 50/70 bitumen were measured and compared with extracted bitumen from different days aged RAP. The Short-Term Ageing (STA) of the asphalt mixtures was carried out at 135 °C for 4 h as per SHRP-A-383 [24]. After STA,

the mixtures were subjected to a Long-Term Ageing (LTA), the temperature for the long-term ageing was considered 95 °C, which was according to the report of the National Cooperative Highway Research Program (NCHRP) [25]. For this purpose, the asphalt mixtures were placed on a tray with the size of (250 × 350 × 50) mm with a maximum layer thickness of 30 mm and maintained in the oven at 95 °C for 120 h after the STA. This novel ageing technique was utilized because it can deliver a more accurate simulation of the ageing happening in the viscoelastic component of the asphalt mixtures i.e., the binder, and thus, provide an enhanced understanding of the oxidative ageing mechanisms. As the literature has shown an oxidative ageing mechanism is one of the major causes of deterioration in asphalt pavement. So, for this reason, in this study, we have considered such an ageing protocol to investigate the deterioration of asphalt mixtures. The effect of traffic loading could not be considered in the study due to the limitations of laboratory ageing methods and due to the limited existing knowledge to implement such factors on the laboratory scale. The same procedure was applied at each ageing cycle for producing the RAP of the recycled mixtures of the next recycling cycles. At the end of the ageing procedure, the aged binder was extracted by standard binder extraction and recovery method. Tetrachloroethylene was the solvent used for the extraction, and the distillation was done by means of a rotavapor.

### 3.4. Binder Ageing

The properties of bitumen 50/70 were considered as the reference during the optimization of the rejuvenator and the recycling process. As indicated in Figure 2, in order to have the aged reference properties, a widely accepted protocol that simulates short-term ageing and long-term ageing was also considered. The short-term ageing was performed using a Rolling Thin film Oven (RTFO) according to EN 12607-1:2014 [26]. For this purpose, 35 g of bitumen was weighed and placed in glass bottles, where it was kept in an RTFO setup at a temperature of 163 °C for 75 min with an airflow rate of 4 litres/minute and the rotation of the carriage assembly at a rate of 15 rotations/minute. Afterward, the long-term ageing of bitumen was carried out, using a pressure ageing vessel (PAV) according to EN 14769:2012 [27]. For this purpose, 50 g of STA binder was placed on a circular tray and kept in the PAV at a conditioning temperature of 100 °C with a pressure of 2.1 MPa for 20 h.

### 3.5. Multiple Recycling

The multi-recycling of the asphalt mixtures containing 50% RAP was simulated by using multiple ageing and recycling of the mixtures by applying a loose mixture ageing technique. During each cycle of asphalt mixture recycling, 50% RAP from the prior cycle and 50% virgin materials (unaged) were used. It is worth mentioning that since the study was developed at a binder scale, the extracted binders were recycled 3 cycles after each cycle of mixture ageing.

The required properties of the mixtures were achieved using the rejuvenator optimized at each cycle of recycling. The rejuvenator was added to the hot unaged (virgin) bitumen at 160 °C on the weight of the aged binder content. The asphalt binder blend was then stirred manually for approximately 1 min. Finally, to provide consistency and homogeneity the blend was kept in the oven for 10 min. The dosage of the rejuvenator was optimized using the Binder Fast Characterization Test/BTSV test [22]. The description of the BTSV test is detailed in the dedicated section afterwards. In addition, the dosage was additionally controlled by other tests such as standard penetration and softening point temperature, as a commonly used approach for the optimization of the rejuvenators, as well [28]. For this purpose, initially, different dosages of rejuvenator were added to the extracted aged binders, and the blends were tested and compared with the reference bitumen values. Table 3 shows the optimized dosages of the rejuvenator used at each cycle of recycling.

**Table 3.** The optimum dosage of the rejuvenator for each cycle of recycling.

| Binders from the Control Mix | Cycle of Rejuvenation | Optimum Dosage * (%) | Binders from the Testing Mix | Cycle of Rejuvenation | Optimum Dosage * (%) |
|---|---|---|---|---|---|
| $CA^0$ + R | 1 | 4.5 | $GRPA^0$ + R | 1 | 5.5 |
| $CA^1$ + R | 2 | 4.5 | $GRPA^1$ + R | 2 | 4.5 |
| $CA^2$ + R | 3 | 4.4 | $GRPA^2$ + R | 3 | 4.4 |

* The dosage of rejuvenator is on the weight of the aged binder.

### 3.6. Testing Methods

According to the objectives of this research, the experimental plan shown in Figure 2 consisted of a full binder scale characterization by means of both conventional tests and complementarily rheological analysis using a Dynamic Shear Rheometer (DSR). In addition to the physical and rheological analysis, the binders were chemically characterized using Fourier Transform Infrared (FTIR) spectroscopy. All the tests were carried out according to the European standard testing methods. A detailed description of all the tests is provided in the sections dedicated to each test, the results, and the analysis.

## 4. Results and Discussion

### 4.1. Penetration Test

According to the standard testing method EN 1426:2015 [29], the penetration test determines the consistency and the deformation of any asphalt binders. In the present study, an automatic digital penetrometer from the manufacturer Infratest has been used. The penetration depth of the needle is determined with the electronic position measuring system equipped with the device. During the penetration test, the average of five readings was considered to determine the penetration values, which are shown in Figure 5. According to the results of the penetration test, the binders extracted from both the long-term aged control asphalt mixture Mix–$CA^0$, and the long-term aged GRP modified mixture Mix–$GRPA^0$, have shown similar penetration values. These values were also similar to the reference aged (RTFO + PAV) Bit. 50/70. Although, during the first cycle of rejuvenation, different dosages were used (optimized), still it can be seen that the penetration values of the extracted binder from GRP-modified asphalt mixtures are comparable with the binder extracted from the control mixture and the reference Bit. 50/70.

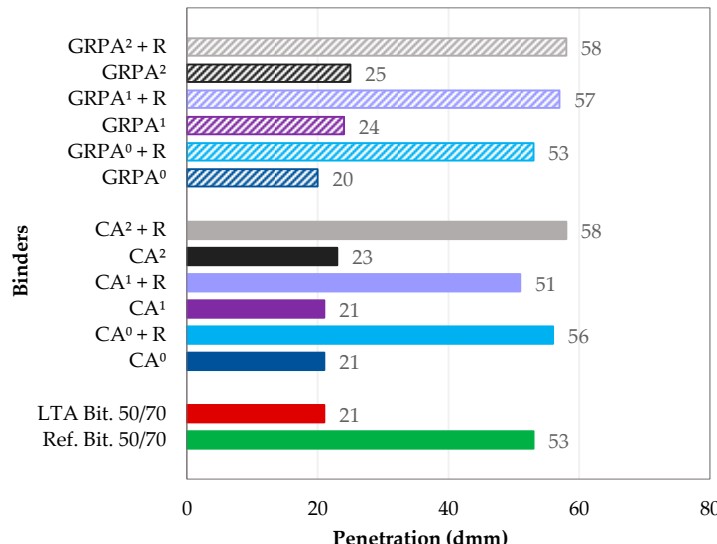

**Figure 5.** Penetration values.

In the further cycles of ageing and recycling, the dosages used for rejuvenations were the same and the obtained values were comparatively similar to the extracted binders from

the control mixture and reference Bit. 50/70. Having such values, it can be stated except for the first-round results that are impacted by the modification, the extracted binders of the further cycles are similar. The result of the penetration test conducted on all the rejuvenated blends at every recycling stage complied with the required penetration range of 50–70 dmm recommended by the Italian technical specification [21].

### 4.2. Softening Point Test

The softening point test is used to determine the thermal sensitivity of bitumen or an asphalt binder. The test is performed as per EN 1427:2015 [30] on an automatic ring and ball tester from the manufacturer Infratest.

The results of the softening point test are summarized in Table 4. Considering the obtained values, the extracted binder from the long-term aged GRP-modified asphalt mixture, Mix–$GRPA^0$, showed a slightly higher softening point compared to the extracted binder from the control mixture, $CA^0$, and the LTA Bit. 50/70. The slight increase in the softening point of $GRPA^0$ was expected due to the addition of the GRP additive. It is noteworthy that the higher softening point was also recorded for the binder extracted from the GRP-modified asphalt mixture immediately after the production and prior to the ageing phase, where 55.3 °C was obtained compared to 51.7 °C for the control mixture. In fact, the addition of GRP additive enhances higher resistance to permanent deformation, which was identified by the increase in softening point before the ageing phase. Moreover, considering the results after long-term ageing showed that the inclusion of GRP additive has a slight impact on the ageing mechanism compared to the $CA^0$ and the LTA Bit. 50/70. Considering the target of this research to bring back the properties to the original one, at the first cycle of recycling comparatively higher dosage of rejuvenator, shown in Table 3 was used for the binder $GRPA^0$. Accordingly, during the further phases of ageing and recycling, the used rejuvenator dosage (optimized dosage) and softening point values of the extracted binders were comparable. From another point of view, it was noted that the mixtures containing rejuvenator from the last cycle showed less susceptibility to ageing. This can be seen where for both sets of the extracted binders, the softening point values are less than the cycle before. According to the Italian technical specification, the softening point for a bitumen 50/70 should be in the range of 45 to 60 °C [21]. Therefore, the results of all the rejuvenated blends showed within the range of the specification limit.

**Table 4.** Softening points of the binders.

| Reference Binders | Softening Point (°C) | Binders from the Control Mix | Softening Point (°C) | Binders from the Testing Mix | Softening Point (°C) |
|---|---|---|---|---|---|
| LTA Bit. 50/70 | 60.8 | $CA^0$ | 60.8 | $GRPA^0$ | 62.5 |
| Ref. Bit. 50/70 | 49.6 | $CA^0$ + R | 49.7 | $GRPA^0$ + R | 50.2 |
| LTA Bit. 50/70 | 60.8 | $CA^1$ | 60.6 | $GRPA^1$ | 59.4 |
| Ref. Bit. 50/70 | 49.6 | $CA^1$ + R | 50.3 | $GRPA^1$ + R | 49.3 |
| LTA Bit. 50/70 | 60.8 | $CA^2$ | 58.6 | $GRPA^2$ | 58.5 |
| Ref. Bit. 50/70 | 49.6 | $CA^2$ + R | 49.2 | $GRPA^2$ + R | 48.8 |

### 4.3. Dynamic Viscosity Test

The viscosity of the asphalt binder is a property that defines its resistance to flow and deformation. In this study, a Brookfield rotational viscometer was used to determine the dynamic viscosity of the asphalt binder at specific temperatures following EN 13302:2018 [31]. The test examines the torque needed to rotate the cylindrical spindle submerged in the binder at a constant speed by using a thermoset temperature system. The torque is later converted into viscosity, where the viscosity is a temperature-dependent property. The viscosity of asphalt binder lowers at an increasing rate of temperature.

According to the results, shown in Figure 6, the extracted binders of both sets of testing and control asphalt mixtures showed the same trend after each ageing and recycling cycle. The trend recorded was very similar to the penetration and softening point tests. It is noteworthy that as was expected due to the presence of GRP, the binder extracted from the first round aged modified mixture, $GRPA^0$ showed a higher viscosity compared to the control mixture. However, after the first cycle of recycling the extracted binder from both control and GRP-modified mixtures performed comparatively and the same optimum dosage of the rejuvenator was optimized for recycling the aged binders at recycling cycles 2 and 3.

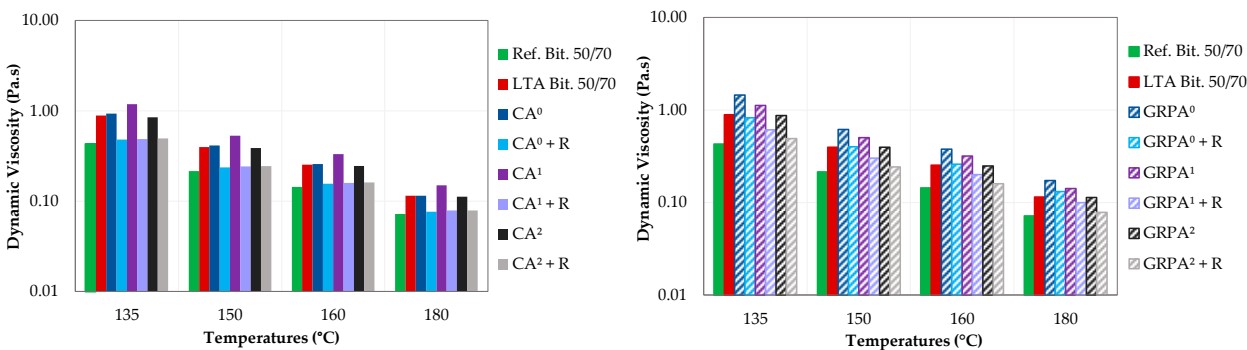

**Figure 6.** Dynamic viscosity: on the left, binders from the control asphalt mix, and on the right, binders from the GRP-modified asphalt mix.

### 4.4. Dynamic Shear Rheometer (DSR) Analysis

The Anton Paar MCR 102 Dynamic Shear Rheometer (DSR) was used in the study to determine the rheological properties of the bituminous binders in accordance with EN 14770:2012. The test parameters include complex shear modulus, G\*, and phase angle, δ. The complex shear modulus, G\* comprised of two components, the dynamic storage modulus G′ and the loss modulus G″. The phase angle, δ between the sinusoidal load or deformation and the material's response, as well as the amplitudes of shear stress, $\tau_0$, and shear strain, $\gamma_0$, are used to calculate G′ and G″ [32]. During the DSR analysis, several testing parameters were investigated and compared for the reference and testing asphalt binders. Upper PG, G\*, phase angle, loss factor, and equi–shear modulus temperature and its corresponding phase angle were the parameters considered in the analysis and presented in the following section. In this research PG grading and temperature sweep was carried out in accordance with EN 14770:2012 [33].

#### 4.4.1. Temperature Sweep Test

The temperature sweep test was carried out using the test settings as follows:

- Plate geometry: 25 mm with a 1 mm gap;
- Test temperature range: 25 °C to 90 °C;
- Temperature increment rate: 1 °C/min;
- Constant shear stress: 500 ± 5 Pa;
- Test frequency: 1.59 Hz (10 rad/s).

The test results presented in Figures 7 and 8 are in terms of Loss factor (tan δ) and Isochronal plots of G\*. The loss factor of a material depends on the loss modulus, G″ and storage modulus, G′. The G″ is a measure of the material's lost deformation energy and it contributes to the viscous part of the material and whereas, the G′ is a measure of energy stored in the system, which reflects the elastic part of the material. The loss factor usually represents material losses, which is an indication of the softness or hardness of a material [34]. The parameter, G″ is directly proportional to the loss factor, where an increase in G″ accordingly increases the loss factor and their divergence indicates the gel–sol transition. The rheological analysis conducted using such a technique is considered

valid for the study of the behaviour of ageing and rejuvenation of asphalt binders since G′–G″ equilibrium represents the structure of a material. The tan δ can be expressed as the ratio of G″ to G′.

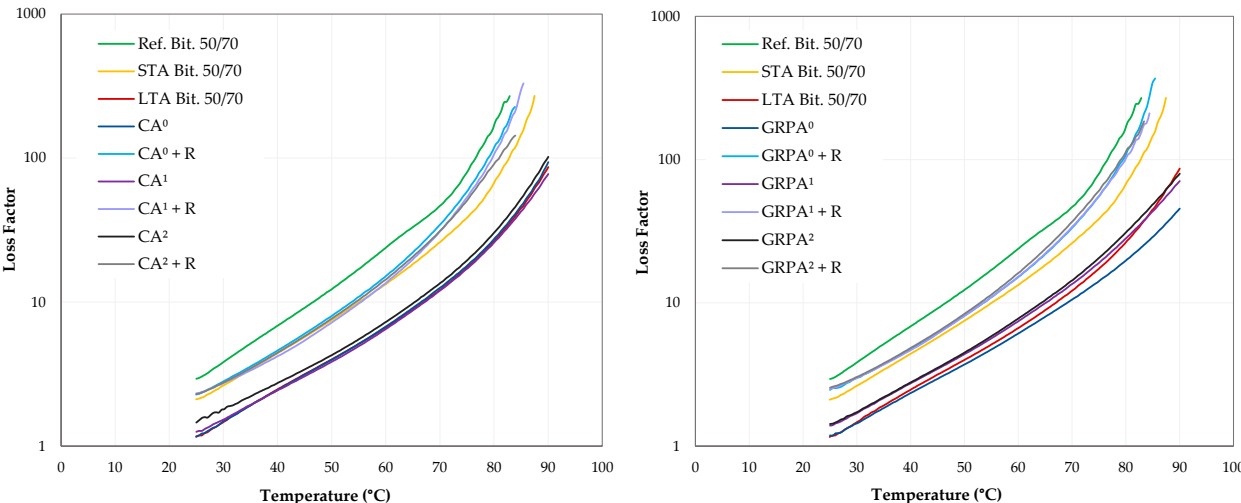

**Figure 7.** Loss factor versus temperature curve: on the left, binders from the control asphalt mix, and on the right, binders from the GRP-modified asphalt mix.

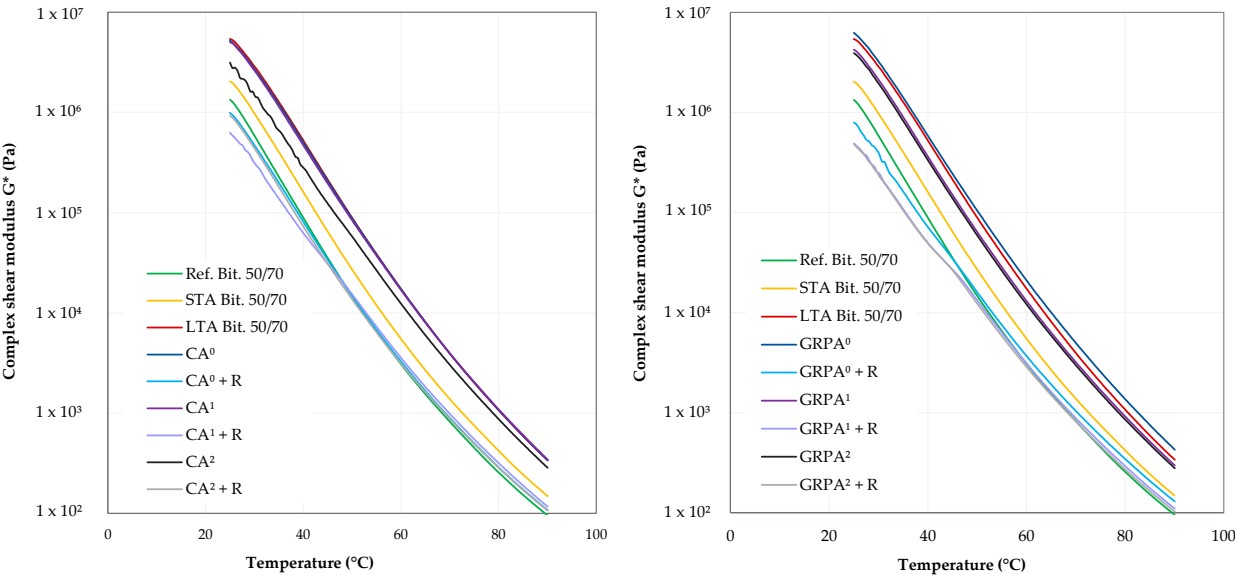

**Figure 8.** Isochronal plots: on the left, binders from the control asphalt mix, and on the right, binders from the GRP-modified asphalt mix.

According to the obtained results, the following points can be drawn:

- As expected, both short-term and long-term ageing increased the stiffness of the reference Bit. 50/70, however, within the loss factor figure, the significance of short-term ageing is more visible compared to other testing methods.
- None of the recycled binders, neither the control mix binders nor the GRP mix binders, were shown the same loss factor plots compared to the Ref. Bit. 50/70 virgin bitumen. It is worth mentioning that if a higher dosage of rejuvenator had been added during the recycling cycles, the binders would become excessively soft, and the corresponding mixtures would be susceptible to rutting. This fact is also attested by the loss factors at high temperatures, where the recycled binders showed similar loss factors at these temperatures.

- It is noteworthy that despite the difference between the recycled binders' loss factor curves and Ref. Bit. 50/70 at unaged (virgin) state, the aged binders showed very similar curves after long-term ageing with the PAV aged bitumen 50/70 (LTA Bit. 50/70).
- Regardless of the Ref. Bit. 50/70, as it was observed via other testing methods, the extracted binders from the control and the GRP-modified asphalt mixtures after the first cycle of recycling showed comparative performance.
- Not similar to the loss factor plots, the G* isochronal plots of the extracted binders from the aged/recycled control and testing asphalt mixtures were comparable to the reference bitumen 50/70 in its un-aged state. This could assure the accuracy of the optimized rejuvenator dosage.

### 4.4.2. Performance Grading

The upper-performance grading of the samples was tested completing/confirming the performance of the binders at high temperatures. The test results are summarized and compared in Table 5. According to the obtained results, the upper PGs complied with the trend obtained via softening points and loss factors. The same as the other tests' results, the PG values show that the presence of the GRP increased the upper PG value as a modifier. At each recycling cycle, the addition of the rejuvenator reduced the PG of the binders extracted from the control and the testing mixture to the target value of reference bitumen 50/70.

**Table 5.** Upper-performance grading.

| Reference Binder | Upper PG (°C) | Upper Critical Temp. (°C) | Binders from the Control Mix | Upper PG (°C) | Upper Critical Temp. (°C) | Binders from the Testing Mix | Upper PG (°C) | Upper Critical Temp. (°C) |
|---|---|---|---|---|---|---|---|---|
| LTA Bit. 50/70 | 76 | 80 | $CA^0$ | 76 | 80 | $GRPA^0$ | 82 | 82.2 |
| Ref. Bit. 50/70 | 64 | 67.5 | $CA^0$ + R | 64 | 68.5 | $GRPA^0$ + R | 70 | 70.6 |
| LTA Bit. 50/70 | 76 | 80 | $CA^1$ | 76 | 80.1 | $GRPA^1$ | 76 | 78.4 |
| Ref. Bit. 50/70 | 64 | 67.5 | $CA^1$ + R | 64 | 69.4 | $GRPA^1$ + R | 64 | 68.0 |
| LTA Bit. 50/70 | 76 | 80 | $CA^2$ | 76 | 78.2 | $GRPA^2$ | 76 | 78.4 |
| Ref. Bit. 50/70 | 64 | 67.5 | $CA^2$ + R | 64 | 68.3 | $GRPA^2$ + R | 48.8 | 68.2 |

### 4.4.3. BTSV Test

According to EN 17643:2022 [35], the Binder Fast Characterization Test or BTSV (Bitumen-Typisierungs-Schnell-Verfahren) test was used to determine the equi–shear modulus temperature, $T_{BTSV}$ at which the bituminous binder exhibits a complex shear modulus of 15 kPa under a stress-controlled oscillation mode in a non-stationary temperature state and to determine the corresponding phase angle $\delta_{BTSV}$ without the need for viscoelastic domain. The test involves a steady increase in the test temperature within the range of 25 °C to 90 °C, while the specimen was subjected to a constant shear stress of 500 ± 5 Pa under the oscillation at a frequency of 1.59 Hz. Nevertheless, in the scenario where there is no measured value within the range of complex shear modulus, G* = 15 ± 0.05 kPa, $T_{BTSV}$ and $\delta_{BTSV}$ are calculated by interpolating nearest values using an exponential value for $T_{BTSV}$ and a linear function for $\delta_{BTSV}$.

The BTSV test results in terms of equi–shear temperature and phase angle are summarized in Table 6. As can be seen, not only the equi–shear temperatures but also the respective phase angles obtained at the optimum dosage of the rejuvenator, were comparable. This could also confirm the quality and right dosage of the rejuvenator in recovering the rheological parameters that have been considered also during the rejuvenator optimization in this study. Moreover, the obtained equi–shear modulus temperature of all the binders was similar to their corresponding softening point values.

**Table 6.** BTSV results.

| Reference Binder | $T_{BTSV}$ (°C) | $\delta_{BTSV}$ (°) | Binders from the Control Mix | $T_{BTSV}$ (°C) | $\delta_{BTSV}$ (°) | Binders from the Testing Mix | $T_{BTSV}$ (°C) | $\delta_{BTSV}$ (°) |
|---|---|---|---|---|---|---|---|---|
| LTA Bit. 50/70 | 61.0 | 81.9 | $CA^0$ | 60.9 | 82 | $GRPA^0$ | 62.2 | 81.7 |
| Ref. Bit. 50/70 | 49.9 | 85.3 | $CA^0$ + R | 49.9 | 82.8 | $GRPA^0$ + R | 50.4 | 83.2 |
| LTA Bit. 50/70 | 61.0 | 81.9 | $CA^1$ | 60.8 | 81.7 | $GRPA^1$ | 59.1 | 81.9 |
| Ref. Bit. 50/70 | 49.9 | 85.3 | $CA^1$ + R | 50.1 | 82.2 | $GRPA^1$ + R | 49.2 | 82.7 |
| LTA Bit. 50/70 | 61.0 | 81.9 | $CA^2$ | 58.7 | 81.6 | $GRPA^2$ | 58.6 | 82 |
| Ref. Bit. 50/70 | 49.9 | 85.3 | $CA^2$ + R | 49.4 | 82.3 | $GRPA^2$ + R | 48.7 | 82.7 |

*4.5. Fourier Transform Infrared (FTIR) Spectroscopy*

FTIR spectroscopy examines the infrared light absorbed by bonds in molecules when it has the same frequency as the vibration frequency of the bonds, enabling the identification of different chemical functionalities based on the wavenumber. The analysis of the spectrum is carried out considering the intensity of the peaks, which represents the concentration of the bonds or functional groups. The wavelength of the spectral peaks corresponding to the Sulfoxide group (around 1030 cm$^{-1}$) and the Carbonyl group (around 1700 cm$^{-1}$) are considered [32,36]. The spectral peaks corresponding to the wavelength around 1460 cm$^{-1}$ and 1376 cm$^{-1}$ are considered as the baseline for the peak analysis. These spectral peaks are referred to as aliphatic groups (CH$_2$ − CH$_3$) and they are not affected significantly during the oxidation process [37].

In the present study, a Bruker Tenson 27 FTIR spectrometer with an Attenuated Total Reflectance (ATR) diamond crystal was used to investigate the chemical changes that occur due to the ageing and rejuvenation in an asphalt binder. The test was conducted by placing approximately 10 mg of asphalt binder directly onto the diamond crystal and the spectra were recorded within the wavelength range of 600–4000 cm$^{-1}$ with a resolution of 4 cm$^{-1}$ and an average of 32 repetitive scans. The analysis of the spectrum was carried out by calculating the Sulfoxide index and Carbonyl index, using the formula below [38], and the calculated values are presented in Table 7.

$$\text{Sulfoxide Index(SI)} = \frac{Area\ of\ the\ sulfoxide\ band\ centred\ at\ 1030\ \text{cm}^{-1}}{Area\ of\ the\ CH_2\ \text{centred at}\ 1460\ \text{cm}^{-1} + Area\ of\ the\ CH_3\ \text{centred at}\ 1376\ \text{cm}^{-1}} \quad (1)$$

$$\text{Carbonyl Index(CI)} = \frac{Area\ of\ the\ carbonyl\ band\ centred\ at\ 1700\ \text{cm}^{-1}}{Area\ of\ the\ CH_2\ \text{centred at}\ 1460\ \text{cm}^{-1} + Area\ of\ the\ CH_3\ \text{centred at}\ 1376\ \text{cm}^{-1}} \quad (2)$$

The spectra of all binders are shown in Figures 9 and 10. As it has been shown in the literature, comparing the spectra of the reference bitumen unaged and LTA aged, it can be clearly seen how the oxidation is represented by increasing the peaks of the sulfoxide and carbonyl band. This phenomenon was also observed for the binders of both control and testing mixtures to different extents. Adding the rejuvenators at each cycle of recycling could also restore some of the chemical properties, especially at the first cycle of recycling where both oxidation representative bands were reduced compared to the original state. However, the same trend was not observed during all the recycling cycles. This could be due to firstly, the chemical components and nature of the rejuvenators and the recycled plastic modifier and secondly, due to the dosage of the rejuvenator, which has been reported in many research works. Moreover, from another point of view, the significance of rejuvenation impact was not the same for both sets of binders, still, the same trend can be observed considering each cycle of ageing and rejuvenation. Overall, it is worth mentioning that for further information and quantifying the impact of the recycled plastic modifier, particularly the dry method, a further chemical investigation should be

carried out i.e., different dosages of the asphalt modifier, which was out of the scope of this research.

**Table 7.** Calculated values of sulfoxide and carbonyl indices.

| Binders | Sulfoxide Index (SI) | Carbonyl Index (CI) |
|:---:|:---:|:---:|
| Ref. Bit. 50/70 | 0.58 | 0.36 |
| LTA Bit. 50/70 | 0.16 | 0.08 |
| $CA^0$ | 0.60 | 1.13 |
| $CA^0 + R$ | 0.15 | 0.08 |
| $CA^1$ | 0.36 | 0.19 |
| $CA^1 + R$ | 0.31 | 0.26 |
| $CA^2$ | 0.29 | 0.59 |
| $CA^2 + R$ | 0.25 | 0.62 |
| $GRPA^0$ | 0.36 | 0.26 |
| $GRPA^0 + R$ | 0.33 | 0.21 |
| $GRPA^1$ | 0.31 | 0.37 |
| $GRPA^1 + R$ | 0.19 | 0.41 |
| $GRPA^2$ | 0.23 | 0.32 |
| $GRPA^2 + R$ | 0.23 | 0.34 |

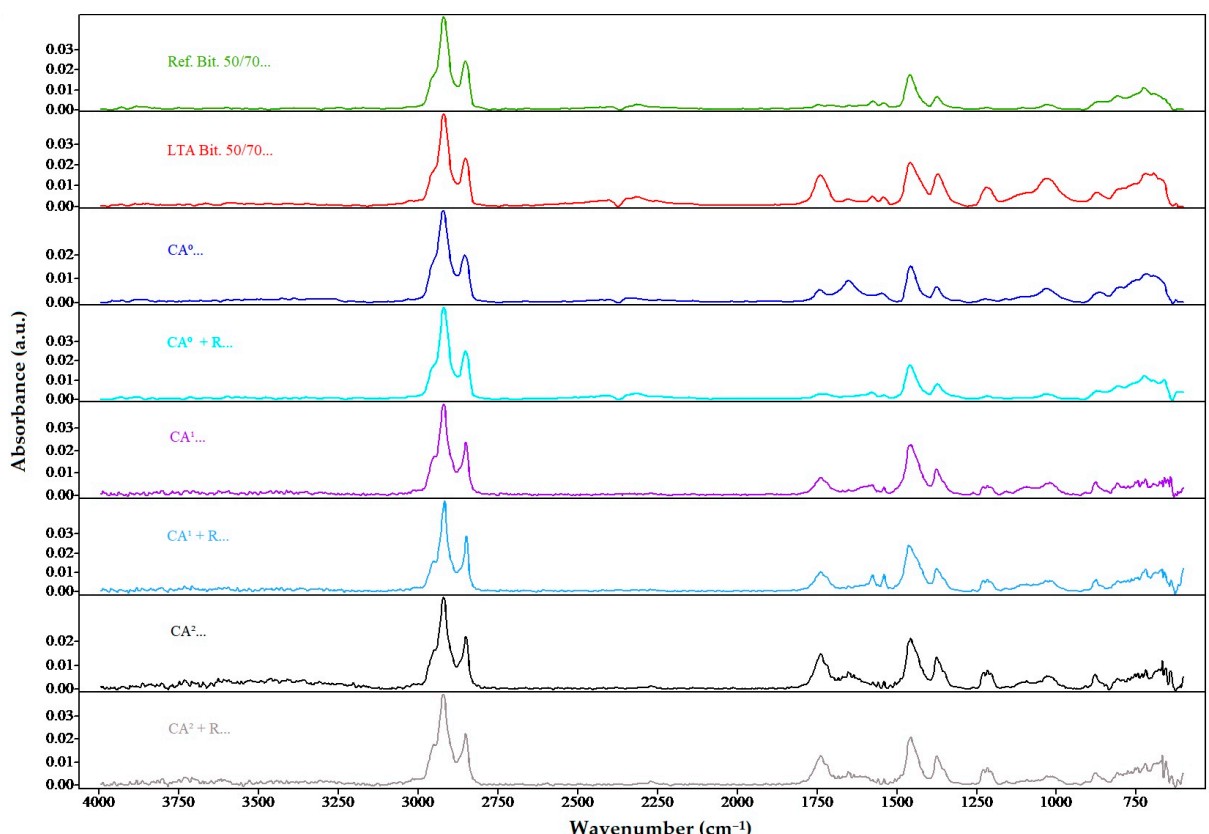

**Figure 9.** Average obtained spectra of binders extracted from the control asphalt mixtures.

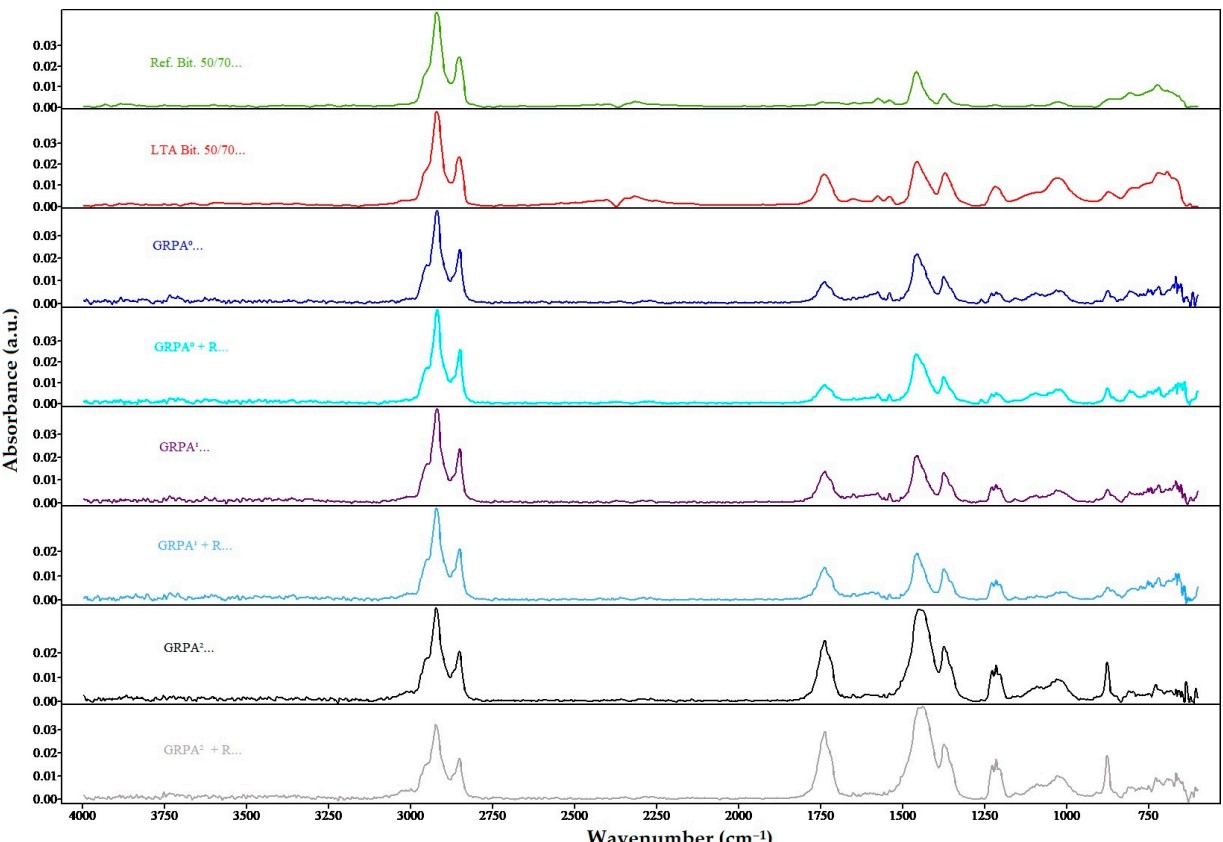

**Figure 10.** Average obtained spectra of binders extracted from the GRP-modified asphalt mixtures.

## 5. Conclusions

In this study, the potential multi-recyclability of asphalt mixtures was investigated via a binder-scale characterization, where three sets of data were compared and analysed. These three sets comprised (i) a reference bitumen 50/70, (ii) the extracted binders from a control mixture without any asphalt modifier, and (iii) the extracted binders from a mixture containing recycled plastic asphalt modifier. In this investigation, the recycling rate was considered 50% and the mixtures were designed and optimized based on the Italian specifications for a dense graded wearing course. The study was structured so as to include 3 recycling cycles of the aforementioned materials. In this case, analysing the obtained results, it can be seen that as a modifier the recycled plastic asphalt additive increases the softening point, upper PG, and equi–shear modulus temperature. However, in this study, it should be considered that since the method of modification was the dry method (process) the asphalt mixture mixing and manufacturing process could also impact the mentioned properties. Moreover, at the first cycle of recycling, a slightly higher dosage of rejuvenator was found to be optimal for the binder extracted from first-cycle aged GRP-modified asphalt mixtures. This was because the modifier enhanced the properties of the mixture and in such case bringing back to the targeted unmodified state of the binder, the dosage needed to be considerably higher than the normal case. During the following cycles, the same dosages of rejuvenator were found as optimal for both control and testing mixtures. Furthermore, the rheological analysis has shown that optimizing the rejuvenator is a critical factor in asphalt recycling, especially for mixtures with high RAP content. The non-optimal dosage can lead to an adverse effect on the properties of the asphalt binder and the produced asphalt mixture. Accordingly, a complete rejuvenator optimization should comprise the properties of bitumen from different aspects and different temperatures. In addition to the physical and rheological properties, the extracted binders with and without a modifier, at different cycles of ageing and rejuvenation, were chemically characterized by means of FTIR spectroscopy. Although, the technique provided a clear image of the ageing impact on the

reference bitumen, the same impact was not observed at all three cycles of rejuvenation for the extracted binders, the influence of the rejuvenator was clearly observed by a reduction in the carbonyl and sulfoxide bands. Finally, the binders corresponding to the reclaimed asphalt containing the introduced recycled plastic modifier showed similar/comparable properties to the binders of the control mixtures without the asphalt modifier. This has been recorded at all three recycling cycles of the extracted binders. Overall, through the obtained results and analysis, it is possible to immediately understand that the partial substitution of virgin materials with recycled ones, can provide end products that have similar, or in some cases depending on the investigated property, better performance than the conventional ones.

Now, bearing in mind the importance of the sustainability and circular economy implications of the pavement engineering sector, the relevance of this study becomes obvious. In fact, the pavement engineering industry exploits tremendous amounts of natural, non-renewable resources during the extraction of the materials, their transport, the production of binders, and of course asphalt mixtures. The impacts even extend up to the construction of pavements and eventually their end of life. Keeping this in mind and in order to shift towards a more circular and sustainable way of conducting business, the potential multi-recycling of materials is of paramount importance. It is worth reiterating, that in this study a novel ageing technique was utilized because it can deliver a more accurate simulation of the ageing happening in the viscoelastic component of the asphalt mixtures, i.e., the binder, and thus, provide an enhanced understanding of the ageing mechanisms. Hence, overall, it is proven that for the asphalt binders investigated, it is possible to expect at least three life cycles of service life with a performance that falls well inside the acceptable windows of the performance of conventional binders. In other words, any asphalt binder can have a tremendous potential for multi-recyclability given that the optimal amount and appropriate type of rejuvenator are selected and used according to a proper rheological characterisation. This paradigm can enable a plethora of opportunities for the involved stakeholders and the relevant decision-makers towards the actual implementation of the circular economy by cutting down on the usage of virgin materials, transport costs, and distances, and allowing a product to be present in the market for longer period of times without losing its value or performance. Obviously, this is a lab-scale investigation, proving the ability to recycle conventional and plastic-modified asphalt mixtures for at least three life cycles based on binder scale investigation without any mechanical performance degradation and before the actual full-scale implementation, a mixture-scale investigation along with a multi-recycling progress monitoring tool is also proposed and is underway.

**Author Contributions:** Conceptualization, V.V. and G.D.M.; Methodology, V.V., G.D.M. and S.E.; Software, V.V.; Validation, V.V. and G.D.M.; Formal analysis, V.V. and S.E.; Investigation, V.V.; Resources, G.D.M., K.M. and L.V.; Data curation, V.V., S.E. and K.M.; Writing—original draft preparation, V.V.; Writing—review and editing, V.V., G.D.M., S.E., K.M. and L.V.; Visualization, V.V., G.D.M. and K.M.; Supervision, G.D.M. and K.M.; Project administration, G.D.M. All authors have read and agreed to the published version of the manuscript.

**Funding:** This research received no external funding.

**Data Availability Statement:** It will be available upon request.

**Acknowledgments:** The authors would like to acknowledge Iterchimica S.p.A., Sicilbitumi S.r.l., and Cava Giardinello S.r.l. for their collaboration and availability, and also for providing the testing materials for this study.

**Conflicts of Interest:** The authors declare no conflict of interest.

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
