# Peer review of "Investigating the Multi-Recyclability of Recycled Plastic-Modified Asphalt Mixtures"

_infrastructures, doi:10.3390/infrastructures8050084_

Round 1

Reviewer 1 Report

This manuscript covers a laboratory investigation on the multi-recyclability of recycled plastic-modified asphalt mixtures. Asphalt mixtures containing 50% RAP with and without a recycled plastic asphalt modifier and rejuvenating agent were studied. The recycled plastic asphalt modifier was made of hard recycled plastics and was introduced to the mixture via a dry method. The manuscript has been carefully reviewed and has been found to be well written. The study has presented interesting and relevant findings on the potential multi-recyclability of asphalt mixtures. Find below for your consideration minor comments with request for clarifications and/ or further improvement of the manuscript:

1.      In Line 127-128, it is stated that “It is worth mentioning that the waste plastics used to produce this asphalt modifier are hard plastics and not softer ones like plastic bags and water bottles” It would be good if you can mention the specific recycled plastics e.g. PET, PE-HD, PVC etc.

2.      In Line 167-169, it is stated that “The choice of as 50% RAP incorporation in the study was based on the foreseen level of recycling in asphalt industry of Italy and the feasibility of recycling approach in many existing asphalt plants”. This not clear and more justification is required. There has to be a scientific consideration rather than targeted future expectations.

3.      In Figure 2. “Research plan”, there are many abbreviations used that have not been first provided in full when first presented in the manuscript.

4.      RAP was produced in this study by artificial aging following a laboratory ageing protocol. As you clearly pointed out, in situ aged RAP is vital in any recycling study since it also incorporates materials deterioration as a function of traffic loading and environmental conditions. How is the effect of traffic loading on ageing of RAP considered in the laboratory produced RAP?

5.      In Line 331-332, it is stated that “This higher performance with GRP additive in fact would allow a higher resistance to permanent deformation to the mixture”. Is this increase in softening point of Mix-GRPA⁰ compared with CA⁰ and the LTA Bit. 50/70 of about 1.7⁰ considered significant from a practical point of view?

6.      Test results on the parameters such as penetration, softening point, dynamic viscosity etc., should be compared with the acceptable specification requirements to confirm that they comply with the national and/or international specifications (e.g., Italian technical specification).

7.      It would be good to provide figures showing the laboratory test setup for the tests conducted.

Reviewer 2 Report

This paper conducted extensive experiments on various ways of high RAP incorporation. This is a lab-scale investigation, proving the ability to recycle conventional and plastic modified asphalt mixtures for at least 3 life cycles based on binder scale investigation without any mechanical performance degradation. The results can enable a plethora of opportunities for the involved stakeholders and the relevant decision-makers towards the actual implementation of circular.

Reviewer 3 Report

The  article  investigating the multi-recyclability of recycled plastic-modified asphalt mixtures,there are some suggestions as follows:

1.Line 292,” the needle penetration “ is unprofessional terminology, please amend.

2.Line 293-298, this is common sense knowledge and is recommended to be removed or rewritten.

3.Line 291, Line 316, Line 355,  Line 377, Line 483, the technological equipment on which the experiments were carried out and the type of factory equipment must be specified for any methods and tests. 

4.Line 317-322, It is not recommended to write the test procedure for the softening point, please modify it.

5.It is proposed to streamline the content of Discussion & Conclusions.

6.Too many experimental processes and steps are written in the article, and it is recommended to modify. For example Line 355.

Reviewer 4 Report

Thanks for preparing a paper for review and publication.  This is a very timely project.  A better description of the test plan and associated test results are needed.  Test results need to be better summarized.
